# Aphid Resistance Evaluation and Constitutive Resistance Analysis of Eighteen Lilies

**DOI:** 10.3390/insects14120936

**Published:** 2023-12-08

**Authors:** Huajin Shi, Jian Zhong, Yilin Liang, Peng Zhang, Liuyu Guo, Chen Wang, Yuchao Tang, Yufan Lu, Ming Sun

**Affiliations:** State Key Laboratory of Efficient Production of Forest Resources, Beijing Key Laboratory of Ornamental Plants Germplasm Innovation and Molecular Breeding, National Engineering Research Center for Floriculture, Beijing Laboratory of Urban and Rural Ecological Environment, Key Laboratory of Genetics and Breeding in Forest Trees and Ornamental Plants of Ministry of Education, School of Landscape Architecture, Beijing Forestry University, Beijing 100083, China; shihuajinbjfu@163.com (H.S.); zhongjianbjfu@163.com (J.Z.); elaineleung@bjfu.edu.cn (Y.L.); gzyyndemailbox@163.com (P.Z.); guo1727805874@163.com (L.G.); wangchenbjfu@163.com (C.W.); tangyuchao@bjfu.edu.cn (Y.T.); lyf1997@bjfu.edu.cn (Y.L.)

**Keywords:** lily, *Aphis gossypii*, aphid resistance, palisade tissue

## Abstract

**Simple Summary:**

*Aphis gossypii* is an important pest that harms lilies and limits the development of the lily industry. Improving host plant resistance is an effective and environmentally friendly method for aphid control. We studied the resistance of 16 lily cultivars and 2 wild lily species to *A. gossypii* and the biological characteristics of their leaves. Among the 18 tested lily plants, ‘Palazzo’, ‘Nymph’, ‘Cameleon’, and *Lilium lancifolium* showed strong resistance. The analysis of the correlation between the thickness of leaf palisade tissue and the number of greenhouse aphids reveals a significant negative correlation. This indicates that the thicker palisade tissue may be responsible for the observed resistance. Identifying these lilies is important for managing aphid populations and provides additional solutions for lily-integrated pest management.

**Abstract:**

Lilies (*Lilium* spp.) are famous bulb flowers worldwide, with high ornamental value. Aphid damage has seriously constrained the development of the lily industry. In this study, the aphid resistance of 16 lily cultivars and 2 wild lily species was characterized in the field and greenhouse. Leaf color parameters, stomatal density and size, thickness of leaf layers, leaf waxy content, and leaf water content were determined to explore the constitutive resistance of lilies. The results show that there was a significant positive correlation between the number of aphids in the field and in the greenhouse (*p* ≤ 0.05, r = 0.47). This indicated that the level of aphid infestation in both the field and the greenhouse is generally consistent across different types of lily plants. Among these 18 lilies, ‘Palazzo’, ‘Nymph’, ‘Cameleon’ and *L. lancifolium* were resistant to *A. gossypii*, while ‘Black Beauty’ and ‘Magnefique’ had poor resistance. The correlation analysis results showed that the number of aphids was negatively correlated with leaf abaxial surface a*, stomatal size, water content, and thickness of leaf palisade tissue and positively correlated with leaf distal axial surface b*, C*, and waxy content. Among them, the correlation between the number of aphids and the thickness of leaf palisade tissue reached a significant level (*p* ≤ 0.05, r = −0.521). This indicated that the thickness of the palisade tissue of lily leaves might be an important factor influencing the proliferation of aphids. This study not only screened out aphid-resistant lilies but also established a crucial research foundation for the targeted breeding and molecular breeding of lilies with aphid resistance.

## 1. Introduction

Insect infestation has always been an important limiting factor for green crop production. With the large-scale development of crop production, the threat of pests has rapidly developed from a few species causing a small amount harm within a small area to large-scale damage being inflicted by multiple species with a high degree of harm [1]. Among them, aphids have become a severe pest, affecting the development of production and the improvement of efficiency. Aphids often cluster on the young leaves and buds of plants, causing serious damage through nutrient feeding, honeydew deposition, and viral disease transmission [2]. This infestation leads to plants shriveling, affecting flowering and fruiting, and reducing ornamental and application value, resulting in serious economic losses [3]. The main method of aphid management is the use of pesticides, which not only increases production costs but also causes harm to the environment [4,5,6,7]. Therefore, people are beginning to search for alternative biological control methods for pesticides. Enhancing host plant defenses through breeding is the most sustainable, effective, and eco-friendly method [8].

Plants have their own defense systems that are developed during long-term interactions with insects. These defense mechanisms can be classified into two categories: constitutive resistance and induced resistance [4,9,10]. Constitutive resistance is inherent in plants and is related to their genotypes [11,12]. It acts throughout their lifespan and consists of physical barriers and chemical barriers [4,10,13]. Research on physical barriers in plants currently focuses on factors such as leaf color, leaf trichomes, the wax content of the leaf surface, leaf structure, and so on [14,15]. For example, many phytophagous insects show a preference for a yellow-green color and have no tendency for a red color [16,17,18]. Resistant cultivars usually have longer and denser trichomes, as well as increased leaf cuticle thickness, palisade tissue thickness, and spongy tissue thickness [15,19,20].

Lilies (*Lilium* spp.) are important perennial herbaceous bulbous plants that are primarily distributed in the temperate regions of the Northern Hemisphere, including Eastern Asia, Europe, and North America [21]. Lilies are cultivated as ornamental plants worldwide, serving various purposes such as cut flowers, potted flowers, and urban landscaping [22]. Due to their beautiful flowers and pleasant fragrance, lilies occupy a significant portion of the market in the global cut flower industry and are highly appreciated by customers [23]. Additionally, the steroidal saponins, flavonoids, and polysaccharides contained in the *Lilium* are the main active substances with pharmacological activities, such as anti-tumor, anti-inflammatory, and antioxidant activities [21]. Thus, lilies are also widely cultivated as edible and medicinal plants in several Asian countries [22,24]

Biotic stress caused by pests and pathogens seriously hinders the sustainable development of the lily industry. Among the pests that harm lilies, cotton aphids are particularly problematic. They tend to gather on the tender leaves and buds of lilies to suck their juices, which poses a significant threat to the plant’s health and ornamental value. In addition, cotton aphids can cause sooty blotch and spread viruses such as lily mosaic virus (LMV) and lily ring spot virus (LRSV) [3,25,26]. Despite the threat posed by cotton aphids, there are very few studies related to aphid resistance in lilies. Currently, only a few lily cultivars have been studied for their resistance to aphids, and little is known about the mechanisms related to aphid resistance in lilies [3]. In this study, the aphid resistance of eighteen lilies was evaluated, then the aphid-resistant mechanisms were systematically explored by means of morphology, physiology, and metabolite analysis. This study aimed to screen out aphid-resistant lily germplasms directly and lays an important research foundation for targeted breeding and molecular breeding. Ultimately, the cultivation of new aphid-resistant lily cultivars is expected to reduce the use of pesticides, lower production costs, and contribute to the green ecological sustainability of the lily industry.

## 2. Materials and Methods

### 2.1. Plant Preparation

The 18 lily plant materials for testing were planted in the National Engineering Research Center for Floriculture (China, Beijing, Changping District, 116.446° E, 40.151° N) (Table 1). Before planting, the lily bulbs were soaked in carbendazim solution at 500 mg/L for 40 min, then the rotting and diseased scales were peeled off, and the basal roots were pruned to 1 cm. The greenhouse lilies were grown in pots with single bulbs (diameter of pots: 17 cm, height: 15 cm). The cultivation substrate was a mixture of peat/perlite/vermiculite at a ratio of 5:1:1 with carbendazim added for sterilization. When planting, we kept the bud tip of the lily bulbs upward, and the substrate was 5~8 cm above the top of the lily bulbs.

Field planting was carried out in early April. The treated lily bulbs were planted in the experimental field according to the completely randomized zone group method, with the burial depth consistent with that of potted plants. Finally, we watered the bulbs thoroughly to ensure that they were tightly bonded to the soil.

### 2.2. Aphid Rear

The aphids, identified as *A. gossypii*, were harvested from lilies at the National Engineering Research Center, in Beijing, China (Appendix A). The morphological identification was based on the Journal of Economic Insects of China and the Ecological Atlas of Aphids in Beijing [27,28]. *A. gossypii* used in the aphid resistance experiment were reared on the lily cultivar ‘Magnefique’ at 22 ± 2 °C and 65 ± 5% relative humidity with a photoperiod of 16:8 (light: dark) in the breeding greenhouse.

### 2.3. Aphid Resistance Test

#### 2.3.1. Greenhouse Test

Five fourth-instar *A. gossypi* apterous nymphs were placed on the young leaves at the top of each lily stem and confined with a voile bag (20 × 30 cm, mesh = 80) when the lilies grew to a height of 10~15 cm [29] (Appendix A). The number of *A. gossypii* was counted after 10 days. The environmental conditions were kept the same as in Section 2.2. Five biological replicates were set. No insecticides were used during the experiment.

#### 2.3.2. Field Test

When the lilies in the field grew to a height of 10–15 cm, they were manually inoculated with aphids for resistance evaluation tests. Each plant was inoculated with five fourth-instar *A. gossypi* apterous nymphs, and the number of *A. gossypi* was counted after 10 days, with 5 biological replicates for each plant material. It was ensured that no insecticides were applied around the test field during the test period.

### 2.4. Determination of Biological Parameters of Lily Leaf

#### 2.4.1. Leaf Preparation

The fifth and sixth leaves under the lily buds were collected to test when the lilies in the greenhouse grew to a height of 10–15 cm.

#### 2.4.2. Leaf Color

To determine L* (the lightness), a* (red to green axis), and b* (yellow to blue axis) of the leaf’s adaxial and abaxial surfaces, we used a spectrophotometer (NF555, Nippon Denshoku, Japan) to measure the middle part. We followed a previous study’s measuring method [18]. Five biological replicates were set. Leaf color was analyzed using two color spaces, CIELAB (L*a*b*) and CIE L*C*h*, selected based on their wide acceptance by the industry and scientific community. C* (Chroma) and h* (Hue angle) were calculated using the following formula:C* = (a*^2^ + b*^2^)^1/2^
h* = arctan(b*/a*)

#### 2.4.3. Leaf Stomata

Leaf stomata were investigated by using the imprinting method referenced from a previous study [30]. Colorless nail polish was applied to the middle of the lower epidermis of the lily fresh leaves, and gently torn off after the nail polish dried to make a clinical slide. These slides were then observed and photographed with an optical microscope (Sdptop CX40P, Sungrant, Suzhou, China). For each cultivar or species, three samples were selected, and for each sample, five random visual fields were observed. The number of stomata, stomatal length, and stomatal width of each field were counted with Image-Pro plus 6.0 (Media cybernetics, Rockville, MD, USA). Stomatal density, which is the number of stomata per unit leaf area (square millimeters), was calculated.

#### 2.4.4. Leaf Anatomical Structure

The lily fresh leaves were cut crosswise into 0.5 cm-wide slices at the center and fixed with FAA fixative (Solarbio G2350) for 70 h. The fixed material was dehydrated in ethanol, made transparent in xylene, embedded in paraffin, and then sliced using a microtome (Leica RM 2016, Wetzlar, Germany) with a section thickness of 12 μm. Safranine O-fast green staining was performed after the specimen was dried. Then, it was sealed with neutral resin and photographed with an optical microscope (Sdptop CX40P, Sungrant, Suzhou, China). Three samples were selected for each plant material, and each sample was randomly observed in three fields of view. Data were measured by Image-Pro plus 6.0 (Media cybernetics, Rockville, MD, USA). The measured parameters included the thickness of the leaf, upper epidermis, palisade tissue, spongy tissue, and lower epidermis. The method of making paraffin sections and the measurement standard of data can be found in a previous study [31].

#### 2.4.5. Leaf Waxy Content

Leaf waxy content measurement was referenced and improved from a previous study [32]. Fresh lily leaves weighing 2 g were cut and placed into a beaker, where they were soaked in 30 mL of chloroform for a minute. Next, the liquid was filtered and transferred into a clean beaker that had been weighed beforehand. The beaker was then weighed again until all the chloroform had evaporated. Three replicates of the experiment were carried out. The waxy content of the leaves was calculated using the following formulas:

Leaf waxy content (mg/g) = (W2 − W1) × 1000/2.0 g (W1 is the mass of the clean beaker, W2 is the mass of the beaker after chloroform evaporation).

#### 2.4.6. Leaf Water Content

Once the lily leaves were clearly separated and tagged, they were quickly weighed to avoid water loss. Then, the fresh leaves were placed in an oven at a temperature of 110 °C for 10 min and then kept at 80 °C until a constant dry weight was obtained [33]. The determination was repeated three times for each plant material. The leaf water content was calculated by using the following formulas:Leaf water content (%) = (fresh weight − dry weight)/fresh weight × 100%

### 2.5. Statistical Analysis

All data were checked for normality and the homogeneity of variance before statistical analysis. The normality of data distributions was tested using the Shapiro–Wilk normality test. The homogeneity of variance between groups was tested using the Levene test for homogeneity of variance. The aphid number, leaf color, leaf thickness, waxy content, and other biological parameters of each lily were tested via one-way ANOVA followed by Duncan’s multiple comparisons test with a 95% confidence interval of the difference. Correlation analysis was performed by means of Pearson correlation coefficient calculation. All statistical analyses were performed on SPSS20.0 (IBM SPSS, Somers, NY, USA). Images were created using Origin software (2023b, MicroCal, East Northampton, MA, USA) and Excel 2016 (Microsoft, Redmond, WA, USA).

## 3. Results and Discussion

### 3.1. Aphid Resistance

After 10 days of inoculation, aphids were found to have significantly increased on ‘Magnefique’ and ‘Black Beauty’ in the greenhouse, while ‘Cameleon’ had the lowest aphid proliferation (Figure 1). These results indicate that ‘Magnefique’ and ‘Black Beauty’ displayed weaker aphid resistance compared with the other 16 plant materials, while cultivars like ‘Cameleon’ demonstrated stronger aphid resistance. The field resistance evaluation results also support this finding. However, there are some inconsistencies regarding the aphid resistance of some plants in the greenhouse and field experiments. For example, ‘Cameleon’ displayed the strongest aphid resistance in the greenhouse, while ‘Apricot Fudge’ had the lowest aphid proliferation in the field (Figure 1). Moreover, greenhouse aphid populations were higher than field aphid populations for the same lily materials. As shown in the figure, many aphids in the greenhouse gather on lilies’ tender leaves and flower buds to absorb the juices (Figure 2). These results are probably because greenhouse conditions are more favorable for aphid growth and reproduction and there is less threat from their natural enemies present in the field [34,35,36].

Although greenhouse cut flower production is their main form of application, lilies are still used in many applications as important flower beds and flower sea plants. Therefore, this study combines the aphid resistance evaluation results of 18 lily plants obtained from greenhouse and field evaluations, which provide higher practical application value for the pest control and cultivation management of these lilies. Correlation analysis showed a significant positive correlation between greenhouse aphid numbers and field aphid numbers (*p* ≤ 0.05, r = 0.47), indicating that the degree of aphid infestation in the field and greenhouse is generally consistent among different lily plants (Figure 3). This laid the foundation for the further exploration of the aphid resistance of lilies.

Different plants have various insect resistance strategies such as the waxiness of the epidermis, stomatal density, and deeper vascular bundle burial depth [10]. However, it is not yet clear why there are significant differences in resistance among various lily plants. After ten days of inoculation, there were notable differences in aphid populations among different lilies. We hypothesize that certain biological properties of lily leaves play a crucial role in hindering the aphids (Figure 1).

### 3.2. Leaf Color

After analysis, significant differences (*p* < 0.05) were found among different lilies in L*, a*, b*, C*, and h* on the adaxial and the abaxial surfaces of leaves. It is well-documented that many phytophagous insects, such as aphids, prefer yellow-green light [37,38]. Consequently, green leaves are more likely to be infested by various insect species as compared to red leaves [16,17]. In this study, ‘Secret Kiss’ had the smallest distal a* value, indicating the most intense green color, and the largest distal b* value, indicating the yellowest color, and it was the least aphid-resistant (Table 2; Figure 1 and Appendix A). In addition, the highly aphid sensitive ‘Magnefique’ also had a very low a* value and high b* value on both the adaxial and abaxial surfaces. The aphid-resistant lilies such as ‘Palazzo’, ‘White Triumph’, and ‘Conca D’or’ exhibited a lower value of brightness (L*), yellowness (b*), and chromaticity (C*) and higher values of redness (a*) on both the adaxial and abaxial surfaces. These results align with the theoretical understanding of aphid visual ecology, which suggests that phytophagous insects may favor specific colors or intensities in their preferred plants [39].

### 3.3. Leaf Stomata

Lilies’ stomatal length, stomatal width, and stomatal density of lilies were found to be significantly different (*p* < 0.05). Previous studies have shown that larger stomatal densities can alter leaf surface topography, leading to behavioral responses from insects [15]. Plant aphid-resistant varieties tend to have a smaller stomatal size and higher stomatal density [30]. The stomatal length and stomatal width of *L. leucanthum*, which is weakly resistant to aphids, were significantly smaller than those of other lilies, and the stomatal density was significantly higher than that of other lilies. The more aphid-resistant ‘Conca D’or’, ‘Palazzo’ had a larger stomatal length and stomatal width, as well as lower stomatal density (Table 3; Figure 1 and Appendix A). This finding is contrary to previous studies, which have suggested that plants with increased stomatal sizes and greater stomatal densities enable the conversion of additional photoassimilates, resulting in strengthened cell walls and enhanced tolerance to aphids [40]. The stomata play a complex role in influencing plant aphid resistance.

### 3.4. Leaf Anatomical Structure

Plants have evolved various constitutive defense systems to protect themselves from insect damage. One such defense mechanism is the presence of palisade tissue and spongy tissue, which act as physical barriers to resist insect attacks [41]. Researchers have previously compared the resistance of cucumbers to aphids and found that the deeper the vascular bundles are buried, the stronger the cucumber’s resistance to aphids. This may be due to the increased difficulty aphids face in feeding on the vascular sap [31]. In this study, significant differences (*p* < 0.05) in the thickness were observed among all the lilies tested. ‘Cameleon’ exhibited the highest leaf thickness, measuring 570.41 μm, and demonstrated greater resistance to aphids. The weakly aphid-resistant ‘Secret Kiss’ had the thinnest palisade tissue at 32.87 μm (Table 4; Figure 1 and Appendix A). There are similar cases in previous studies, like a pear tree study which revealed that leaf thickness and palisade tissue thickness are the key factors influencing aphid feeding [31].

### 3.5. Leaf Waxy Content

Leaf surface wax is a layer of lipophilic compounds that covers the plant surface [42]. It serves to prevent water loss and resist insect attacks. Leaves with a thicker waxy structure have smoother surfaces, which reduces the attachment ability of phytophagous insects and thus leads to lower egg drop and hatchability, resulting in reduced pest damage. A previous study found that removing the wax from the surface of the leaves promoted cochineal insect feeding on *Brassicaceae* [43]. Additionally, the wax content of cabbage was significantly and negatively correlated with green peach aphid preference [36].

However, in this study, the most severely aphid-infested lilies, ‘Magnefique’ and ‘Black Beauty’, had the highest wax content at 1.45 mg/g and 1.44 mg/g, respectively. On the other hand, aphid-resistant lilies, such as ‘Conca D′or’ and ‘Palazzo’, had the lowest wax content of 0.71 mg/g and 0.79 mg/g, respectively (Figure 1 and Figure 4). This anomaly is not exceptional. Another previous study found that epidermal waxes can also benefit plants by influencing natural enemy insects [44]. Lower wax content increases the attachment ability of natural enemy insects such as *Hippodamia convergens* Linnaeus (Ladybirds, Coleoptera), thereby promoting their predation on phytophagous insects [45,46]. These findings highlight the complexity of the role of surface waxes in the tertiary trophic relationship between plants, phytophagous insects, and natural enemy insects.

### 3.6. Leaf Water Content

The water content of lily leaves ranged from 86.02% to 91.67%, with significant differences (*p* < 0.05). Previous studies have shown a positive correlation between host feeding preference and leaf water content [41]. While *L. leucanthum* is susceptible to aphids, its water content was significantly lower than that of the other lilies, at 86.02% (Figure 1 and Figure 5). The highest water content was found in ‘Eyeliner’, ‘Armandale’, at 91.67% and 91.62%, respectively, but they are not aphid-resistant. Previous studies have shown that a decrease in leaf water content and leaf water potential increases the difficulty of aphids to feed in the xylem and shortens the feeding time [47]. The decrease in water potential elevated the total amino acid content of the plant, including asparagine and valine, which are critical for aphid performance. However, aphids did not benefit from improved phloem sap quality [48]. At a lower water content, the expression of JA- and SA-related genes rose in plants due to drought stress, which in turn acted as a suppressor of aphids [49]. The water content of leaves has a more complex relationship with aphid populations.

### 3.7. Correlation between the Aphid Population and Biological Parameters of Lily Leaves

In this study, Pearson’s correlation coefficient was utilized to investigate the relationship between various biological parameters of lilies and their ability to resist aphids. The results indicate that the number of aphids in the greenhouse was significantly negatively correlated with the thickness of the leaf palisade tissue (*p* ≤ 0.05, r = −0.521), suggesting that thicker leaf tissue made it harder for aphids to feed and resulted in a lower aphid population (Figure 6). This finding is consistent with a previous study which showed that the thickness of palisade tissue was one of the main factors influencing pear’s resistance to pear psylla [31]. Additionally, the findings indicate that parameters such as stomatal length, stomatal width, water content, and leaf abaxial surface a* were negatively correlated with the aphid population, while wax content, leaf abaxial surface b*, and C* were positively correlated. Although these correlations did not reach statistical significance, they still suggest that these leaf traits have some influence on aphid resistance in lilies. Previous studies have also reported similar findings [10,37], indicating that a combination of factors may determine the resistance of lilies to aphids.

## 4. Conclusions

As a limitation to the development of the lily industry, *A. gossypii* has attracted much attention from researchers. This study evaluated the aphid resistance of eighteen lilies in the field and greenhouse. The correlation between the number of aphids and lily leaf color parameters, stomatal density and size, thickness of leaf layers, leaf surface wax content, and leaf water content was assessed. The results show that ‘Palazzo’, ‘Nymph’ and *L. lancifolium* displayed higher resistance to *A. gossypii*. In contrast, ‘Black Beauty’ and ‘Magnefique’ were more susceptible to *A. gossypii* than the other lilies. There was a significant negative correlation between greenhouse aphid number and leaf palisade tissue thickness. The resistant lilies ‘Palazzo’, ‘Nymph’, ‘Cameleon’, and *L. lancifolium* have great potential to be used for breeding enhancement and cultivated in the regions where *A. gossypii* is still considered a major concern.

## Figures and Tables

**Figure 1 insects-14-00936-f001:**
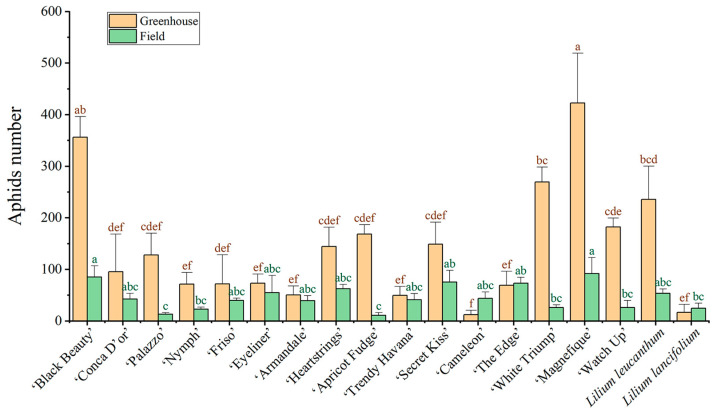
Aphid numbers on eighteen lilies at 10 days after inoculation. Values are the mean ± SE (*n* = 5). Different lowercase letters above the columns indicate significant differences among different lilies at the 0.05 level (Duncan’s test). Greenhouse aphid numbers, F = 6.126; df = 17, 72; *p* < 0.001. Field aphid numbers, F = 2.055; df = 17, 72; *p* = 0.018.

**Figure 2 insects-14-00936-f002:**
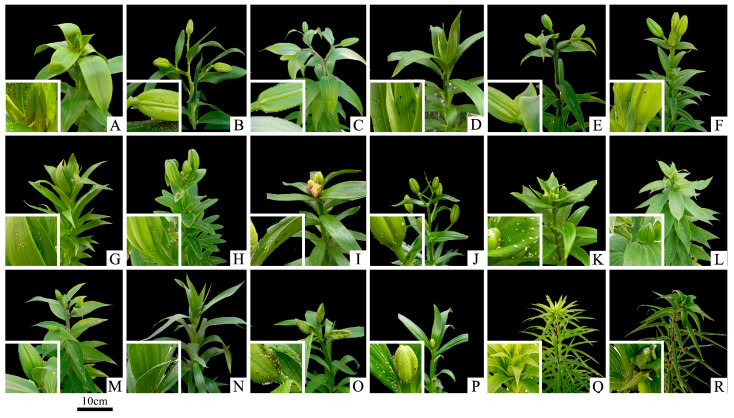
Phenotypes of eighteen lily plants at ten days after inoculation in the greenhouse. The bottom left image shows the top part of the lily magnified three times. (**A**) ‘Black Beauty’; (**B**) ‘Conca D′or’; (**C**) ‘Palazzo’; (**D**) ‘Nymph’; (**E**) ‘Friso’; (**F**) ‘Eyeliner’; (**G**) ‘Armandale’; (**H**) ‘Heartstrings’; (**I**) ‘Apricot Fudge’; (**J**) ‘Trendy Havana’; (**K**) ‘Secret Kiss’; (**L**) ‘Cameleon’; (**M**) ‘The Edge’; (**N**) ‘White Triumph’; (**O**) ‘Magnefique’; (**P**) ‘Watch Up’; (**Q**) *Lilium leucanthum*; (**R**) *Lilium lancifolium*.

**Figure 3 insects-14-00936-f003:**
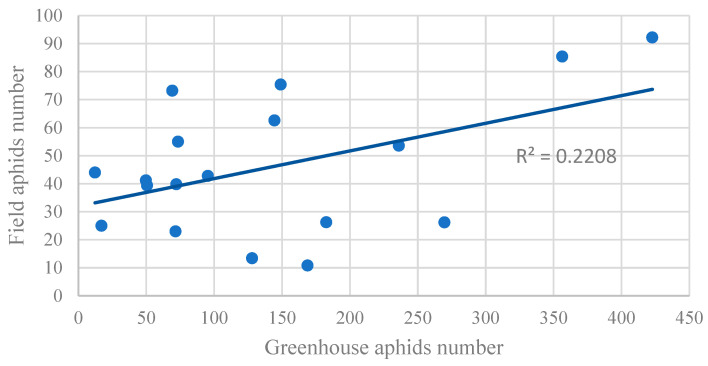
Scatterplot of correlation between greenhouse aphid population and field aphid population (Pearson correlation coefficient).

**Figure 4 insects-14-00936-f004:**
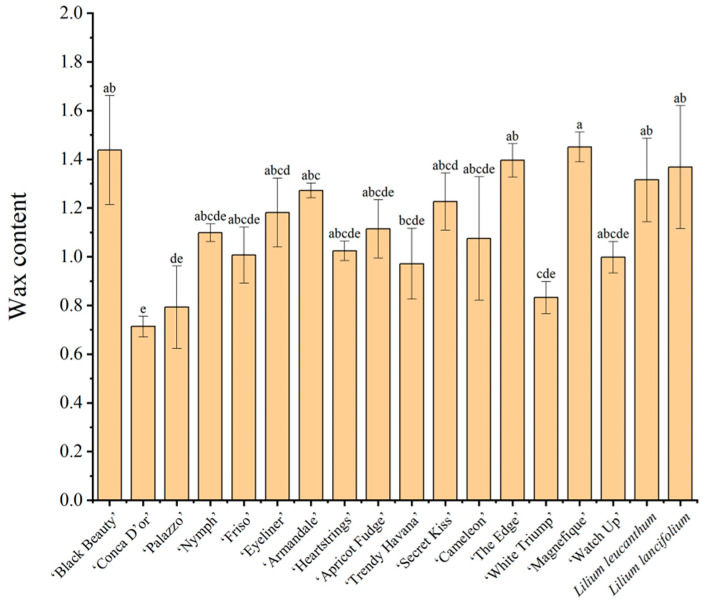
Leaf wax content of eighteen lilies. Values are the mean ± SE (*n* = 3). Different lowercase letters above the columns indicate significant differences among different lilies, Duncan’s test, alpha = 0.05 (F = 2.606; df = 17, 36; *p* = 0.008).

**Figure 5 insects-14-00936-f005:**
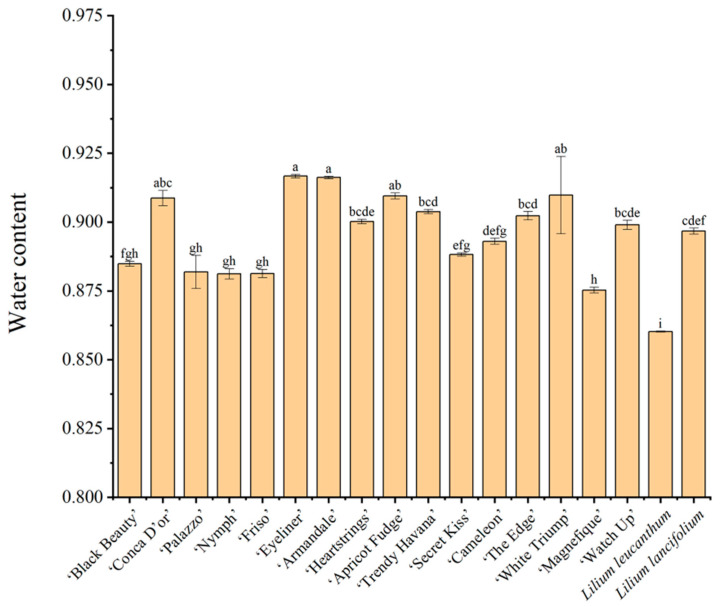
Leaf water content of eighteen lilies. Values are the mean ± SE (*n* = 3). Different lowercase letters above the columns indicate significant differences among different lilies, Duncan’s test, alpha = 0.05 (F = 16.276; df = 17, 36; *p* < 0.001).

**Figure 6 insects-14-00936-f006:**
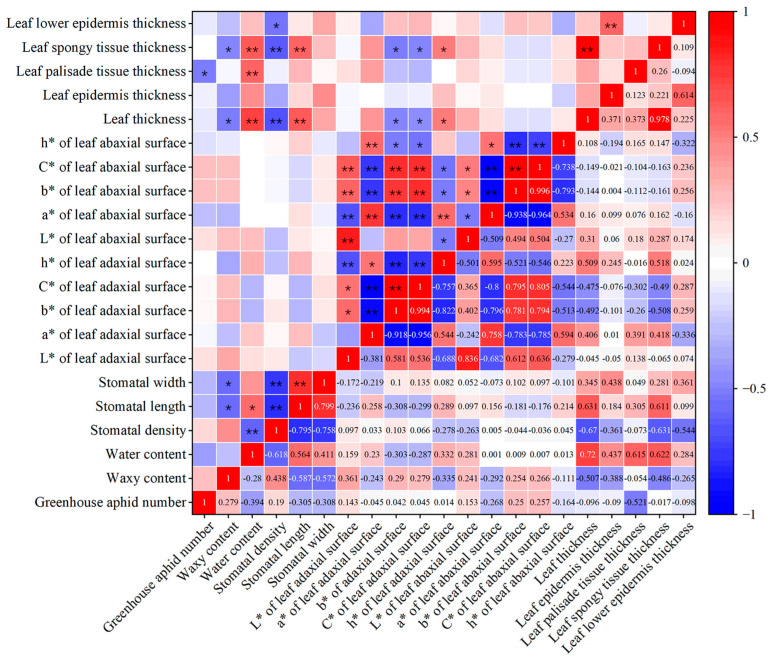
Correlation between the number of greenhouse aphids and biological parameters of lily leaves (Pearson correlation coefficient). Color code ranges from blue = strong negative correlation (r = −1) to white = no correlation (r = 0) to red = positive correlation (r = +1). Labels with “*” indicate the significant correlations at different levels (* *p* ≤ 0.05; ** *p* ≤ 0.01).

**Table 1 insects-14-00936-t001:** Eighteen tested lilies.

ID	Cultivars/Species	Germline
1	‘Black Beauty’	OT
2	‘Conca D′or’	OT
3	‘Palazzo’	OT
4	‘Nymph’	OT
5	‘Friso’	OT
6	‘Eyeliner’	LA
7	‘Armandale’	LA
8	‘Heartstrings’	LA
9	‘Apricot Fudge’	LA
10	‘Trendy Havana’	AA
11	‘Secret Kiss’	AA
12	‘Cameleon’	OO
13	‘The Edge’	OO
14	‘White Triumph’	LO
15	‘Magnefique’	LO
16	‘Watch Up’	LL
17	*Lilium leucanthum*	S
18	*Lilium lancifolium*	S

OT—Oriental hybrids × Trumpet hybrids; LA—*Lilium longiflorum* hybrids × Asiatic hybrids; AA—Asiatic hybrids; O—Oriental hybrids; LO—*L. longiflorum* hybrids × Oriental hybrids; LL—*L. longiflorum* hybrids; S—lily species.

**Table 2 insects-14-00936-t002:** Color parameters of the adaxial and abaxial leaves of eighteen lilies.

Cultivars/Species	Colorimetric Characteristics of the Leaf Apaxial Plane					Colorimetric Characteristics of the Leaf Paraxial Plane				
	L*	a*	b*	C*	h*	L*	a*	b*	C*	h*
‘Black Beauty’	37.04 ± 0.81 gh	−18.16 ± 0.79 efg	23.32 ± 1.29 cde	29.56 ± 1.5 bc	−0.908 ± 0.008 cd	48.3 ± 0.71 def	−15.23 ± 0.23 efg	23.91 ± 0.42 abc	28.35 ± 0.47 ab	−1.006 ± 0.004 efgh
‘Conca D′or’	33.51 ± 0.63 i	−15.31 ± 0.89 bcd	18.54 ± 1.85 fg	24.06 ± 2 de	−0.874 ± 0.019 b	47.23 ± 0.88 ef	−12.37 ± 0.35 ab	18.67 ± 0.71 gh	22.4 ± 0.76 gh	−0.984 ± 0.011 abcde
‘Palazzo’	35.56 ± 1.13 hi	−18.03 ± 0.36 efg	24.44 ± 0.89 bcd	30.37 ± 0.92 bc	−0.934 ± 0.008 def	48.28 ± 0.62 def	−13.93 ± 0.36 cd	20.44 ± 0.77 efg	24.74 ± 0.82 defg	−0.972 ± 0.009 ab
‘Nymph’	35.66 ± 1.3 hi	−16.95 ± 0.29 def	22.28 ± 0.46 def	28 ± 0.53 cd	−0.92 ± 0.004 de	46.46 ± 1.02 f	−13.97 ± 0.23 cd	20.63 ± 0.43 efg	24.91 ± 0.47 def	−0.976 ± 0.006 abcd
‘Friso’	33.19 ± 0.76 i	−15.76 ± 0.42 bcd	19.1 ± 0.95 fg	24.77 ± 1 de	−0.878 ± 0.011 b	47.52 ± 0.4 def	−12.75 ± 0.37 ab	18.8 ± 0.84 fgh	22.72 ± 0.9 fgh	−0.972 ± 0.008 ab
‘Eyeliner’	40.75 ± 0.23 def	−16.67 ± 0.47 cdef	21.93 ± 0.76 def	27.55 ± 0.89	−0.92 ± 0.005 de	49.34 ± 0.81 cde	−15.13 ± 0.55 efg	22.38 ± 1.12 cde	27.02 ± 1.21 bcd	−0.974 ± 0.011 abc
‘Armandale’	45.22 ± 1.14 ab	−18.16 ± 0.53 efg	23.54 ± 1.24 cde	29.74 ± 1.31 bc	−0.912 ± 0.011 cd	52.63 ± 0.51 ab	−14.77 ± 0.25 def	22.86 ± 0.32 bcd	27.22 ± 0.39 bc	−0.996 ± 0.004 cdef
‘Heartstrings’	46.41 ± 0.49 a	−14.85 ± 0.54 bc	23.01 ± 0.8 cde	27.39 ± 0.96 cd	−0.996 ± 0.004 h	52.7 ± 0.21 ab	−14.61 ± 0.24 de	20.9 ± 0.5 def	25.5 ± 0.54 cde	−0.96 ± 0.007 a
‘Apricot Fudge’	40.7 ± 1.08 def	−16.27 ± 0.67 bcde	21.85 ± 1.05 def	27.24 ± 1.24 cde	−0.932 ± 0.007 def	53.04 ± 0.36 a	−14.82 ± 0.2 def	24.51 ± 0.36 abc	28.64 ± 0.4 ab	−1.026 ± 0.002 h
‘Trendy Havana’	41.81 ± 0.45 cde	−19.41 ± 0.55 g	27.32 ± 1.56 ab	33.53 ± 1.59 ab	−0.948 ± 0.014 ef	52.53 ± 0.46 ab	−15.18 ± 0.16 efg	24.72 ± 0.25 ab	29.01 ± 0.3 ab	−1.022 ± 0.002 gh
‘Secret Kiss’	42.82 ± 0.32 bcd	−19.48 ± 0.93 g	29.39 ± 2.09 a	35.27 ± 2.25 a	−0.982 ± 0.013 gh	50.75 ± 0.52 bc	−16.09 ± 0.31 g	25.78 ± 0.62 a	30.39 ± 0.69 a	−1.014 ± 0.002 fgh
‘Cameleon’	37.64 ± 1.11 gh	−18.22 ± 0.54 fg	24.47 ± 1.28 bcd	30.52 ± 1.35 bc	−0.93 ± 0.011 def	46.67 ± 0.17 f	−15.14 ± 0.56 efg	23.7 ± 0.66 abc	28.12 ± 0.86 ab	−1.006 ± 0.005 efgh
‘The Edge’	35.37 ± 0.95 hi	−14.67 ± 0.76 b	17.96 ± 0.96 g	23.19 ± 1.23 e	−0.884 ± 0.002 bc	47.21 ± 0.8 ef	−12.2 ± 0.44 a	17.81 ± 0.42 h	21.59 ± 0.58 h	−0.97 ± 0.009 ab
‘White Triumph’	35.41 ± 0.54 hi	−12.4 ± 0.68 a	13.34 ± 0.78 h	18.21 ± 1.03 f	−0.822 ± 0.008 a	48.49 ± 0.84 def	−13.38 ± 0.35 bc	19.55 ± 0.69 fgh	23.69 ± 0.76 efgh	−0.97 ± 0.007 ab
‘Magnefique’	43.79 ± 0.38 bc	−19.01 ± 0.16 g	26.62 ± 0.43 abc	32.72 ± 0.44 ab	−0.952 ± 0.005 f	52.74 ± 0.27 ab	−15.78 ± 0.25 fg	23.97 ± 0.64 abc	28.7 ± 0.66 ab	−0.99 ± 0.006 bcde
‘Watch Up’	39.1 ± 0.66 fg	−15.24 ± 0.27 bcd	18.49 ± 0.62 fg	23.97 ± 0.64 de	−0.878 ± 0.008 b	49.57 ± 0.97 cd	−13.34 ± 0.4 bc	20.91 ± 0.59 def	24.8 ± 0.71 def	−1 ± 0.005 efg
*Lilium leucanthum*	39.34 ± 0.88 efg	−15.52 ± 0.67 bcd	22.17 ± 1.47 def	27.06 ± 1.58 cde	−0.956 ± 0.011 fg	48.21 ± 1.05 def	−13.21 ± 0.48 abc	20.53 ± 1.2 efg	24.42 ± 1.26 efg	−0.998 ± 0.012 def
*Lilium lancifolium*	38.42 ± 0.66 fg	−14.61 ± 0.49 b	19.93 ± 0.82 efg	24.71 ± 0.94 de	−0.936 ± 0.007 def	51.61 ± 0.45 ab	−13.15 ± 0.33 abc	18.82 ± 0.72 fgh	22.96 ± 0.76 fgh	−0.96 ± 0.009 a
F	23.526	10.971	10.909	10.671	19.107	12.386	11.095	13.044	12.799	8.031
df	17, 72	17, 72	17, 72	17, 72	17, 72	17, 72	17, 72	17, 72	17, 72	17, 72
*p* value	<0.001	<0.001	<0.001	<0.001	<0.001	<0.001	<0.001	<0.001	<0.001	<0.001

Values expressed as mean ± SE (*n* = 5); means in the same column followed by different lowercase letters are significantly different at the 0.05 level (Duncan’ test).

**Table 3 insects-14-00936-t003:** Stomatal parameters of eighteen lilies.

Cultivars/Species	Stomatal Length (μm)	Stomatal Width (μm)	Stomatal Density (Number/mm^2^)
‘Black Beauty’	83.86 ± 2.72 g	56.45 ± 1.35 def	45.73 ± 2.07 cd
‘Conca D′or’	114.24 ± 2.22 b	70.73 ± 1.51 a	24.55 ± 0.33 gh
‘Palazzo’	113.74 ± 1.72 b	70.6 ± 1.94 a	32.93 ± 0.49 f
‘Nymph’	92.64 ± 1.53 ef	59.25 ± 0.57 cd	47.45 ± 1.64 cd
‘Friso’	111.28 ± 2.45 b	63.44 ± 0.7 bc	28.92 ± 1.27 fg
‘Eyeliner’	107.73 ± 4.5 bc	65.18 ± 2.29 b	30.98 ± 0.69 f
‘Armandale’	96.58 ± 2.44 de	59.91 ± 1.76 cd	39.42 ± 0.66 e
‘Heartstrings’	101.25 ± 3.77 cd	58.37 ± 1.08 de	47.13 ± 2.53 cd
‘Apricot Fudge’	125.98 ± 3.48 a	69.94 ± 2.13 a	17.15 ± 0.46 i
‘Trendy Havana’	99.34 ± 3.64 de	70.66 ± 1.8 a	22.43 ± 0.6 h
‘Secret Kiss’	82.33 ± 1.75 g	53.61 ± 0.8 ef	45.43 ± 1.76 cd
‘Cameleon’	81.87 ± 2.67 g	59.5 ± 2.3 cd	70.3 ± 2.47 b
‘The Edge’	85.91 ± 1.87 fg	55.95 ± 1.92 def	44.87 ± 1.14 d
‘White Triumph’	114.96 ± 1.98 b	58.1 ± 0.96 de	39.15 ± 0.96 e
‘Magnefique’	74.57 ± 1.09 h	56.02 ± 0.72 def	50.73 ± 1.07 c
‘Watch Up’	72.38 ± 1.54 h	48.31 ± 0.94 gh	44.82 ± 1.16 d
*Lilium leucanthum*	62.98 ± 1.15 i	47.41 ± 1.24 h	90.12 ± 4.07 a
*Lilium lancifolium*	97.76 ± 1.35 de	52.45 ± 1.19 fg	49.13 ± 2.18 cd
F	47.17	23.90	102.18
df	17, 252	17, 252	17, 252
*p* value	<0.001	<0.001	<0.001

Values expressed as mean ± SE (*n* = 15); means in the same column followed by different lowercase letters are significantly different at the 0.05 level (Duncan’s test).

**Table 4 insects-14-00936-t004:** The thickness of each layer of eighteen lilies.

Cultivars/Species	Leaf Thickness (μm)	Leaf Epidermis Thickness (μm)	Leaf Palisade Tissue Thickness (μm)	Leaf Spongy Tissue Thickness (μm)	Leaf Lower Epidermis Thickness (μm)
‘Black Beauty’	369.29 ± 9.04 ijk	48.85 ± 2.08 bcde	44.56 ± 0.84 cde	243.25 ± 8.65 efgh	30.69 ± 0.97 de
‘Conca D′or’	465.57 ± 21.8 bcdef	73.7 ± 0.72 a	64.67 ± 2.74 abc	284.56 ± 19.11 defg	43.33 ± 2.92 abc
‘Palazzo’	426.97 ± 21.84 defgh	44.71 ± 0.89 cde	42.51 ± 4.95 d	305.15 ± 22.68 cd	32.67 ± 1.73 de
‘Nymph’	369.43 ± 13.06 ijk	41.84 ± 4.12 def	60.81 ± 1.65 abcd	233.41 ± 10.18 gh	36.23 ± 3.17 bcde
‘Friso’	449.7 ± 20.18 cdefg	60.46 ± 1.77 b	68.79 ± 4.26 ab	289.59 ± 18.36 def	37.69 ± 1.88 bcd
‘Eyeliner’	450.53 ± 23.23 cdefg	48.56 ± 7.3 bcde	59.66 ± 3.44 abcd	303.53 ± 15.98 cd	33.08 ± 4.06 de
‘Armandale’	420.45 ± 25.76 efghi	46.89 ± 2.82 cde	66.93 ± 2.57 ab	275.98 ± 23.24 defg	31.42 ± 2.83 de
‘Heartstrings’	470.6 ± 23.65 bcde	78.12 ± 6.78 a	56.42 ± 6.12 bcd	293.41 ± 27.6 de	49.74 ± 1.74 a
‘Apricot Fudge’	346.5 ± 7.63 jk	40.49 ± 4.86 ef	55.88 ± 8.13 bcd	207.19 ± 19.27 h	44.03 ± 4.59 ab
‘Trendy Havana’	406.85 ± 6.72 ghi	75.32 ± 3.85 a	58.64 ± 3.73 bcd	240.2 ± 16.14 fgh	44.4 ± 3.81 ab
‘Secret Kiss’	326.12 ± 13.87 k	32.87 ± 1.58 f	58.01 ± 4.67 bcd	199.12 ± 11.85 h	34.03 ± 3.36 cde
‘Cameleon’	570.41 ± 8.39 a	55.92 ± 2.83 bc	63.9 ± 4.48 abc	414.58 ± 3.76 a	34.15 ± 2.19 cde
‘The Edge’	380.16 ± 11.72 hij	50.54 ± 4.43 bcde	45.62 ± 6.96 cde	238.26 ± 17.55 fgh	39.13 ± 4.28 bcd
‘White Triumph’	519.18 ± 30.57 b	53.82 ± 3.61 bcd	54.37 ± 11.78 bcd	373.21 ± 13.52 ab	38.75 ± 1.2 bcd
‘Magnefique’	500.19 ± 11.74 bc	46.61 ± 0.12 cde	69.4 ± 6.56 ab	345.27 ± 15.33 bc	34.13 ± 1.59 cde
‘Watch Up’	411.08 ± 9.59 fghi	43.06 ± 1.29 def	44.77 ± 3.32 cde	280.61 ± 8.6 defg	35.07 ± 1.64 bcde
*Lilium leucanthum*	478.87 ± 9.07 bcd	45.33 ± 3.5 cde	79.73 ± 13.13 a	321.71 ± 10.18 cd	29.88 ± 3.09 de
*Lilium lancifolium*	266.18 ± 5.89 l	49.68 ± 3.16 bcde	50.2 ± 4.33 bcd	142.59 ± 9.62 i	27.66 ± 2.13 e
F	19.176	11.765	2.741	16.04	4.268
df	17, 36	17, 36	17, 36	17, 36	17, 36
*p* value	<0.001	<0.001	0.005	<0.001	<0.001

Values expressed as mean ± SE (*n* = 3); means in the same column followed by different lowercase letters are significantly different at the 0.05 level (Duncan’s test).

## Data Availability

The datasets in this study are available from the corresponding author upon reasonable request.

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
