# Peer review of "Aphid Resistance Evaluation and Constitutive Resistance Analysis of Eighteen Lilies"

_insects, 2023, doi:10.3390/insects14120936_

Round 1
Reviewer 1 Report
Comments and Suggestions for Authors
The Manuscript [insects-2715270] entitled (Aphid resistance evaluation and constitutive resistance analysis of eighteen lilies) stated a significant positive correlation between the number of Aphis gossypii in the field and in the greenhouse indicating that the degree of aphid infestation in the field and greenhouse is generally consistent among different lily plants. Generally, the experiments are well designed and explained. The manuscript has good results and written very well.
Comments
1- Change title to be [Aphid resistance evaluation and constitutive resistance analysis of eighteen lily cultivars]
2- Line 16: [A. gossypii] instead of [Aphis gossypii]
3- Line 77: delete [and viruses]
4- Line 120: [as previously mentioned in subsection of 2.2.] instead of [as 2.2.]
5- In the text related to Figure 1, 7 and 8: please mention the statistical data [F, df, P values]
6- Line 231: use [cultivars] instead of [species], the same in all text of the manuscript
7- Line 237: Check figure and table numbers, they are incorrect.
8- Table 1 in page 8 is Table 2, correct the numbers of other tables.
9- In the table of page 8: mention the statistical data [F, df, P values], the same comment of tables [Stomatal parameters of eighteen lilies], [Thickness of each layer of eighteen lilies]
10- Line 313: mention Authority, Family and Order of Hippodamia convergens
Author Response
The Manuscript [insects-2715270] entitled (Aphid resistance evaluation and constitutive resistance analysis of eighteen lilies) stated a significant positive correlation between the number of Aphis gossypii in the field and in the greenhouse indicating that the degree of aphid infestation in the field and greenhouse is generally consistent among different lily plants. Generally, the experiments are well designed and explained. The manuscript has good results and written very well.
- Comment:Change title to be [Aphid resistance evaluation and constitutive resistance analysis of eighteen lily cultivars]
Reply: Thank you very much for the kind comment. The lilies used in the test included 16 cultivars and 2 wild species, so we remained the original title “Aphid resistance evaluation and constitutive resistance analysis of eighteen lilies”.
- Comment: Line 16: [A. gossypii] instead of [Aphis gossypii]
Reply: “Aphis gossypii” has been replaced by “A. gossypii”.
- Comment: Line 77: delete [and viruses]
Reply: We have deleted “and viruses” and smoothed the sentences.
- Comment: Line 120: [as previously mentioned in the subsection of 2.2.] instead of [as 2.2.]
Reply: “as previously mentioned in the subsection of 2.2.” has been replaced by “as 2.2.”.
- Comment: In the text related to Figures 1, 7, and 8: Please mention the statistical data [F, df, P values]
Reply: F/df/P values have been mentioned in Figures 1, 7, and 8.
- Comment: Line 231: use [cultivars] instead of [species], the same in all text of the manuscript
Reply: In the sentence, we presented the analysis results of 18 lilies, including 16 cultivars and 2 wild species, so we used “lilies” instead of “species” finally. We have reviewed and revised the entire text to ensure that ' cultivars ',' species', and 'lilies' are used correctly.
- Comment: Line 237: Check figure and table numbers, they are incorrect.
Reply: The figure and table number have been revised in the latest text.
- Comment: Table 1 in page 8 is Table 2, correct the numbers of other tables.
Reply: We sincerely thank you for your careful review and apologize for our carelessness. The table number has been revised in the latest manuscript.
- Comment: In the table of page 8: mention the statistical data [F, df, P values], the same comment of tables [Stomatal parameters of eighteen lilies], [Thickness of each layer of eighteen lilies]
Reply: F/df/P values have been mentioned in the three tables.
- Comment: Line 313: mention Authority, Family and Order of Hippodamia convergens
Reply: Authority, Family, and Order of Hippodamia convergens have been mentioned in the latest manuscript.
Reviewer 2 Report
Comments and Suggestions for Authors
Introduction needs to increase the overview in perspective by including other studies abroad areas where study was developed. Methodology in 2.9 must state programs and line per line of how data was analyzed, the present form is very reduced.
Comments on the Quality of English LanguageEnglish is readable, but some sections like the introduction and results need a revision and reparaphrased lines 56-61, 77-83, 304-312, and 346-350.
Author Response
- Comment: The introduction needs to increase the overview in perspective by including other studies abroad areas where the study was developed.
Reply: Thank you very much for the kind comment. There is very little research on the resistance of lilies to Aphis gossypii, and there are almost no studies abroad. Currently, only two Chinese articles have been found on the identification of lilies' resistance to A. gossypii and we have already cited them. The lack of this field makes this study more meaningful. We hope to receive your understanding.
- Comment: Methodology in 2.9 must state programs and line per line of how data was analyzed, the present form is very reduced.
Reply: We have improved the introduction to data analysis by including specific methods for normality and variance homogeneity testing.
- Comment: English is readable, but some sections like the introduction and results need a revision and reparaphrased lines 56-61, 77-83, 304-312, and 346-350.
Reply: We sincerely thank you for your careful review. We have revised and paraphrased these sections. the detailed content is as follows:
Lines 56-61:
Plants have their own defense systems that are developed during long-term interactions with insects. These defense mechanisms can be classified into two categories: constitutive resistance and induced resistance [4,9,10]. Constitutive resistance is inherent in plants and is related to their genotypes [11,12]. It acts throughout their lifespan and consists of physical barriers and chemical barriers [4,10,13]. Research on physical barriers in plants currently focuses on factors such as leaf color, leaf trichomes, the wax content of the leaf surface, leaf structure, and so on [14,15].
Lines 77-83:
Biotic stress caused by pests and pathogens seriously hinders the sustainable development of the lily industry. Among the pests that harm lilies, cotton aphids are particularly problematic. They tend to gather on the tender leaves and buds of lilies to suck juice, which poses a significant threat to the plant's health and ornamental value. In addition, cotton aphids can cause sooty blotch and spread viruses such as lily mosaic virus (LMV) and lily ring spot virus (LRSV) [3, 25, 26]. Despite the threat posed by cotton aphids, there are very few studies related to aphid resistance in lilies. Currently, only a few lily cultivars have been studied for their resistance to aphids, and little is known about the mechanisms related to aphid resistance in lilies [3]
Lines 304-312:
A previous study found that removing the wax from the surface of the leaves promoted cochineal insect feeding on Brassicaceae [42]. Additionally, the wax content of cabbage was significantly and negatively correlated with green peach aphid preference [36]. However, in this study, the most severely aphid-infested lilies, 'Magnefique' and 'Black Beauty', had the highest wax content at 1.45 mg/g and 1.44 mg/g, respectively. On the other hand, aphid-resistant lilies, such as 'Conca D'or' and 'Palazzo', had the lowest wax content of 0.71 mg/g and 0.79 mg/g, respectively (Figure 7; Figure 1). This anomaly is not exceptional. Another previous study found that epidermal waxes can also benefit plants by influencing natural enemy insects [43].
Lines 346-350:
Additionally, findings indicated that parameters such as stomatal length, stomatal width, water content, and leaf abaxial surface a* were negatively correlated with the aphid population, while wax content, leaf abaxial surface b*, and C* were positively correlated. Although these correlations did not reach statistical significance, they still suggest that these leaf traits have some influence on aphid resistance in lilies. Previous studies have also reported similar findings [10,36], indicating that a combination of factors may determine the resistance of lilies to aphids.
Reviewer 3 Report
Comments and Suggestions for Authors
The paper addresses the critical issue of aphid infestation in lilies, which poses a significant threat to the lily industry. The study aims to evaluate the resistance of different lily plants to Aphis gossypii while examining various leaf characteristics to uncover constitutive resistance mechanisms. The findings indicate that specific lily cultivars, including 'Palazzo,' 'Nymph,' 'Cameleon,' and Lilium lancifolium, exhibit strong resistance to aphids, while others, such as 'Black Beauty' and 'Magnefique,' display lower resistance. Moreover, the thickness of the palisade tissue in lily leaves emerges as a crucial factor influencing aphid proliferation.
To enhance the manuscript's clarity and structure, it is recommended to use subsection headings within the methods section, providing a clear organization of the experimental procedures. Subsections like "Plant Preparation," "Aphid Handling," and each specific measurement or test can be beneficial. Notably, it is important to specify the sample size of aphids used in the assay to ensure result repeatability.
Incorporating figures and tables to illustrate key steps of the experimental setup and highlight essential findings is advisable. A visualized experimental workflow can significantly enhance the reader's understanding. Ensuring consistency in units of measurement and providing instrument precision details, particularly in quantitative data like leaf color measurements, is essential for precision and clarity.
Moreover, it is essential to cite the previous studies that were referenced for the methods. This not only adds credibility to the approach but also enables readers to explore the references for more in-depth understanding.
Regarding Figure 9, which aims to illustrate the correlation between aphid numbers and various biological parameters, there may be alternative visualization options to improve clarity and reader-friendliness. While heatmaps can be valuable for conveying complex relationships between multiple variables, they may become cluttered with a large number of variables or data points. In such cases, consider alternative visualizations like scatter plots or network graphs to effectively display specific relationships between pairs of variables. The choice of visualization should align with the complexity of the dataset and the specific insights you aim to convey.
Comments on the Quality of English LanguageThese are some of the grammar errors and typos that can be found in the manuscript. Careful proofreading and editing can help improve the overall quality of the document.
For example, In the sentence, "significant differences for the aphid resistance of some plants under the greenhouse and field experiments," there should be "significant differences in" instead of "for."
In the sentence, "After 10 days of inoculation, aphids were found to significantly increased on the 'Magnefique' and ‘Black Beauty’," it should be "aphids were found to have significantly increased on 'Magnefique' and 'Black Beauty'."
In the sentence, "While cultivars like 'Cameleon' behaved stronger aphid resistance," it should be "behaved with stronger aphid resistance."
In the sentence, "The field resistance evaluation results also support these findings," it should be "The field resistance evaluation results also support this finding" to maintain subject-verb agreement.
In the sentence, "The lily leaves were cut crosswise, taken 0.5 cm wide in the middle, and fixed with FAA fixative," it should be "were taken" instead of "taken."
In the sentence, "Significant differences (p<0.05) were observed among all tested lilies," it should be "among all the tested lilies" to maintain proper article usage.
In the sentence, "The water content of leaves has a more complex relationship with aphid population," it should be "The water content of leaves has a more complex relationship with aphid populations."
Author Response
- Comment: To enhance the manuscript's clarity and structure, it is recommended to use subsection headings within the methods section, providing a clear organization of the experimental procedures. Subsections like "Plant Preparation," "Aphid Handling," and each specific measurement or test can be beneficial.
Reply: “Plant Preparation”, “Aphid Rear”, “Aphid Resistance Test” and “Determination of Biological Parameters of Lily Leaf” have been used as subsection headings to provide a clear organization of the experimental procedures.
- Comment: Notably, it is important to specify the sample size of aphids used in the assay to ensure result repeatability.
Reply: To identify aphid resistance in both greenhouses and fields, we inoculated each lily plant with 5 aphids and set up 5 replicates for each type of lily. And we have mentioned these in the text.
- Comment: Incorporating figures and tables to illustrate key steps of the experimental setup and highlight essential findings is advisable. A visualized experimental workflow can significantly enhance the reader's understanding.
Reply: We have rearranged the sections and removed duplicate images from the main text, key points were emphasized, and improved clarity.
- Comment: Ensuring consistency in units of measurement and providing instrument precision details, particularly in quantitative data like leaf color measurements, is essential for precision and clarity.
Reply: We have ensured consistency of units and clarified color measurement details. The revised paragraph is as follows:
To determine L* (the lightness), a* (red to green axis), and b* (yellow to blue axis) of the leaf's adaxial and abaxial surfaces, we used a spectrophotometer (NF555, Nippon Denshoku, Japan) to measure the middle part. We followed a previous study's measuring method [18] Five biological replicates were set. Leaf color was analyzed using two color spaces, CIELAB (L*a*b*) and CIE L*C*h*, selected based on their wide acceptance by the industry and scientific community. C* (Chroma) and h* (Hue angle) were calculated using the following formula:
- Comment: Moreover, it is essential to cite the previous studies that were referenced for the methods. This not only adds credibility to the approach but also enables readers to explore the references for a more in-depth understanding.
Reply: Essential references of methods have been cited in this revised manuscript.
- Comment: Regarding Figure 9, which aims to illustrate the correlation between aphid numbers and various biological parameters, there may be alternative visualization options to improve clarity and reader-friendliness. While heatmaps can be valuable for conveying complex relationships between multiple variables, they may become cluttered with a large number of variables or data points. In such cases, consider alternative visualizations like scatter plots or network graphs to effectively display specific relationships between pairs of variables. The choice of visualization should align with the complexity of the dataset and the specific insights you aim to convey.
Reply: Thank you very much for the kind comment. Figure 9 aims to illustrate the correlation between aphid numbers and various biological parameters. Scatter plots can only display the correlation between two parameters. Correlation network graphs cannot display the correlation level and coefficient. Therefore, we think the correlation heatmap is more suitable for this study to display the correlation between aphid numbers and leaf biological parameters.
- Comment: Comments on the Quality of English Language These are some of the grammar errors and typos that can be found in the manuscript. Careful proofreading and editing can help improve the overall quality of the document. For example, In the sentence, "significant differences for the aphid resistance of some plants under the greenhouse and field experiments," there should be "significant differences in" instead of "for."
Reply: Thank you for your careful review. We have conducted proofreading and editing carefully and revised them according to the suggestions of reviewers and the requirements of insects journal. "significant differences for" has been replaced by " significant differences in".
- Comment: In the sentence, "After 10 days of inoculation, aphids were found to significantly increased on the 'Magnefique' and ‘Black Beauty’," it should be "aphids were found to have significantly increased on 'Magnefique' and 'Black Beauty'."
Reply: "have" has been added.
- Comment: In the sentence, "While cultivars like 'Cameleon' behaved stronger aphid resistance," it should be "behaved with stronger aphid resistance."
Reply: "with" has been added.
- Comment: In the sentence, "The field resistance evaluation results also support these findings," it should be "The field resistance evaluation results also support this finding" to maintain subject-verb agreement.
Reply: "these findings" has been replaced by "this finding".
- Comment: In the sentence, "The lily leaves were cut crosswise, taken 0.5 cm wide in the middle, and fixed with FAA fixative," it should be "were taken" instead of "taken."
Reply: The sentence has been rewritten as " The lily leaves were cut crosswise into 0.5 cm-wide slices at the center and fixed with FAA fixative (Solarbio G2350) for 70h. "
- Comment: In the sentence, "Significant differences (p<0.05) were observed among all tested lilies," it should be "among all the tested lilies" to maintain proper article usage.
Reply: "the" has been added to maintain proper article usage.
- Comment: In the sentence, "The water content of leaves has a more complex relationship with aphid population," it should be "The water content of leaves has a more complex relationship with aphid populations."
Reply: "population" has been replaced by "populations".
Round 2
Reviewer 3 Report
Comments and Suggestions for Authors
I am pleased to inform you that after careful consideration, I believe your work demonstrates significant contribution to the field. I see all the questions I brought up have been carefully addressed, and I am pleased to see the necessary corrections and improvements in your English interpretation.